# The Hippo effector Yorkie activates transcription by interacting with a histone methyltransferase complex through Ncoa6

**Yun Qing[1,2†], Feng Yin[1†], Wei Wang[1], Yonggang Zheng[1], Pengfei Guo[1], Frederick Schozer[3], Hua Deng[1], Duojia Pan[1]***

[1]Department of Molecular Biology and Genetics, Howard Hughes Medical Institute, Johns Hopkins University School of Medicine, Baltimore, United States; [2]BCMB Graduate Program, Johns Hopkins University School of Medicine, Baltimore, United States; [3]Department of Biology, Johns Hopkins University, Baltimore, United States

**Abstract** The Hippo signaling pathway regulates tissue growth in *Drosophila* through the transcriptional coactivator Yorkie (Yki). How Yki activates target gene transcription is poorly understood. Here, we identify Nuclear receptor coactivator 6 (Ncoa6), a subunit of the Trithorax-related (Trr) histone H3 lysine 4 (H3K4) methyltransferase complex, as a Yki-binding protein. Like Yki, Ncoa6 and Trr are functionally required for Hippo-mediated growth control and target gene expression. Strikingly, artificial tethering of Ncoa6 to Sd is sufficient to promote tissue growth and Yki target expression even in the absence of Yki, underscoring the importance of Yki-mediated recruitment of Ncoa6 in transcriptional activation. Consistent with the established role for the Trr complex in histone methylation, we show that Yki, Ncoa6, and Trr are required for normal H3K4 methylation at Hippo target genes. These findings shed light on Yki-mediated transcriptional regulation and uncover a potential link between chromatin modification and tissue growth.

***For correspondence:** djpan@jhmi.edu

†These authors contributed equally to this work

## Introduction

The Hippo signaling pathway has recently emerged as a central mechanism in organ size control, tissue regeneration, and stem cell biology (*Harvey and Tapon, 2007*; *Badouel et al., 2009*; *Pan, 2010*; *Zhao et al., 2010*; *Halder and Johnson, 2011*; *Barry and Camargo, 2013*). Initially discovered in *Drosophila* for its critical role in restricting imaginal disc growth, the Hippo pathway comprises several tumor suppressor proteins acting through a core kinase cascade that ultimately phosphorylates and inactivates the transcriptional coactivator Yorkie (Yki) (*Huang et al., 2005*). Consistent with its essential role in normal development and tissue homeostasis, YAP, the mammalian counterpart of Yki, encodes a bona fide oncogene and is overexpressed and/or activated in a wide spectrum of human cancers. Elucidating the molecular mechanism by which Yki functions as a transcriptional coactivator is not only relevant for understanding the fundamental mechanisms of growth control but also has important implications for the development of therapeutic strategies targeting the Hippo pathway in cancer and regenerative medicine.

Posttranslational modifications of histones are important features of transcriptional regulation in all eukaryotes. A particularly prevalent modification involved in transcriptional activation is histone H3 methylation. *Drosophila* contains three COMPASS (complex of proteins associated with Set1)-like histone H3 lysine 4 (H3K4) methyltransferase complexes, each defined by a distinct methyltransferase subunit, namely, Trithorax (Trx), Trithorax-related (Trr), and dSet1 (*Mohan et al., 2011*). Previous genetic

**eLife digest** Cells need to work together for a multi-celled organism, such as a plant or an animal, to thrive. Many complicated signaling pathways therefore exist that allow cells to communicate with one another and to control their own activity in response to the signals that they receive. One such pathway, called the Hippo signaling pathway, regulates when cells grow and divide, which allows organs and tissues to develop correctly and helps to prevent cancerous tumors from forming.

Signaling pathways often control the activity of cells by affecting how particular genes are expressed from DNA. One way of doing this is to activate or inactivate proteins called transcription factors, which bind to sections of DNA to alter the expression of nearby genes. In fruit flies, the Hippo signaling pathway stops cells from dividing by inactivating Yorkie, a protein that binds to and activates certain transcription factors. However, exactly how Yorkie is able to activate these transcription factors was unclear.

For transcription factors to function correctly, they must be able to reach the stretch of DNA where they bind. Therefore, another way to alter gene expression is to change how the DNA is packaged in a cell. This can be done by modifying the proteins that the DNA is wrapped around, which are called histones, for example by using enzymes called methyltransferases to add methyl groups to these proteins.

Qing, Yin et al. looked at the Hippo signaling pathway in the fruit fly *Drosophila*, and found that the Yorkie protein only activates transcription factors when another protein called Ncoa6—which is part of a methyltransferase—binds to it. Furthermore, when the Ncoa6 protein was bound directly to the transcription factor, the tissue grew normally even when the Yorkie protein was not present. These findings reveal the importance of histone modifications in controlling tissue growth, and could provide a new direction in the search for cancer treatments.

analysis has implicated Trx in the maintenance of Hox gene transcription and Trr in ecdysone receptor (EcR)-mediated gene transcription (*Sedkov et al., 2003*). Nuclear receptor coactivator 6 (Ncoa6) is a specific subunit of the Trr complex in both *Drosophila* and mammals (*Mohan et al., 2011*). Although its *in vivo* function remains undefined in *Drosophila*, the mammalian Ncoa6 orthologue (also known as NRC, ASC-2, TRBP, PRIP, and RAP250) is essential for embryonic development (*Kuang et al., 2002*; *Antonson et al., 2003*; *Zhu et al., 2003*; *Mahajan et al., 2004*). The mammalian Ncoa6 has been shown to potentiate the activity of nuclear hormone receptors and other DNA-binding transcription factors, at least in part, by recruiting the H3K4 methyltransferases (*Mahajan and Samuels, 2008*). Interestingly, like YAP, the mammalian Ncoa6 is a pro-survival and anti-apoptotic gene (*Mahajan et al., 2004*) and is amplified in multiple cancer types such as breast, colon, and lung cancers (*Lee et al., 1999*).

In this study, we identify Ncoa6 as a Yki-binding protein that is required for transcriptional regulation by the Hippo signaling pathway. We provide evidence showing that the transcriptional coactivator function of Yki depends on its ability to interact with Ncoa6 and that the Trr methyltransferase complex is functionally required for Hippo-mediated growth and gene expression. We further show that Yki, Ncoa6, and Trr are required for normal H3K4 methylation at Hippo target genes. Thus, Yki functions as a transcriptional coactivator, at least in part, by recruiting a H3K4 methyltransferase and altering the chromatin state of target genes.

## Results and discussion

We recently reported a genome-wide RNAi screen in *Drosophila* S2R+ cells using a luciferase reporter driven by a minimal Hippo Responsive Element (HRE) from the Hippo target gene *diap1* (*Koontz et al., 2013*). Briefly, *Drosophila* S2R+ cells were transfected with Yki- and Sd-expressing vectors, together with HRE-luciferase reporter and Pol III-Renilla expression vector as an internal control. Transfected cells were then seeded into individual dsRNA-containing 96-well plates. After RNAi depletion, HRE-luciferase reporter activity was measured and normalized to the Renilla control. This RNAi screen allowed us to uncover both positive and negative regulators of the HRE-luciferase reporter. We have previously characterized a negative regulator from the screen, Tgi (*Koontz et al., 2013*). Here, we focus on the positive transcriptional regulators (*Figure 1—figure supplement1*).

One of the positive regulators identified in our primary screen is Nuclear receptor coactivator 6 (Ncoa6), which was confirmed by repeating the RNAi experiment in triplicate using re-synthesized dsRNA (*Figure 1—figure supplement1*). We were particularly interested in Ncoa6 since it contains three PPxY sequences (*Figure 1A*), which represent a well-established ligand binding motif for

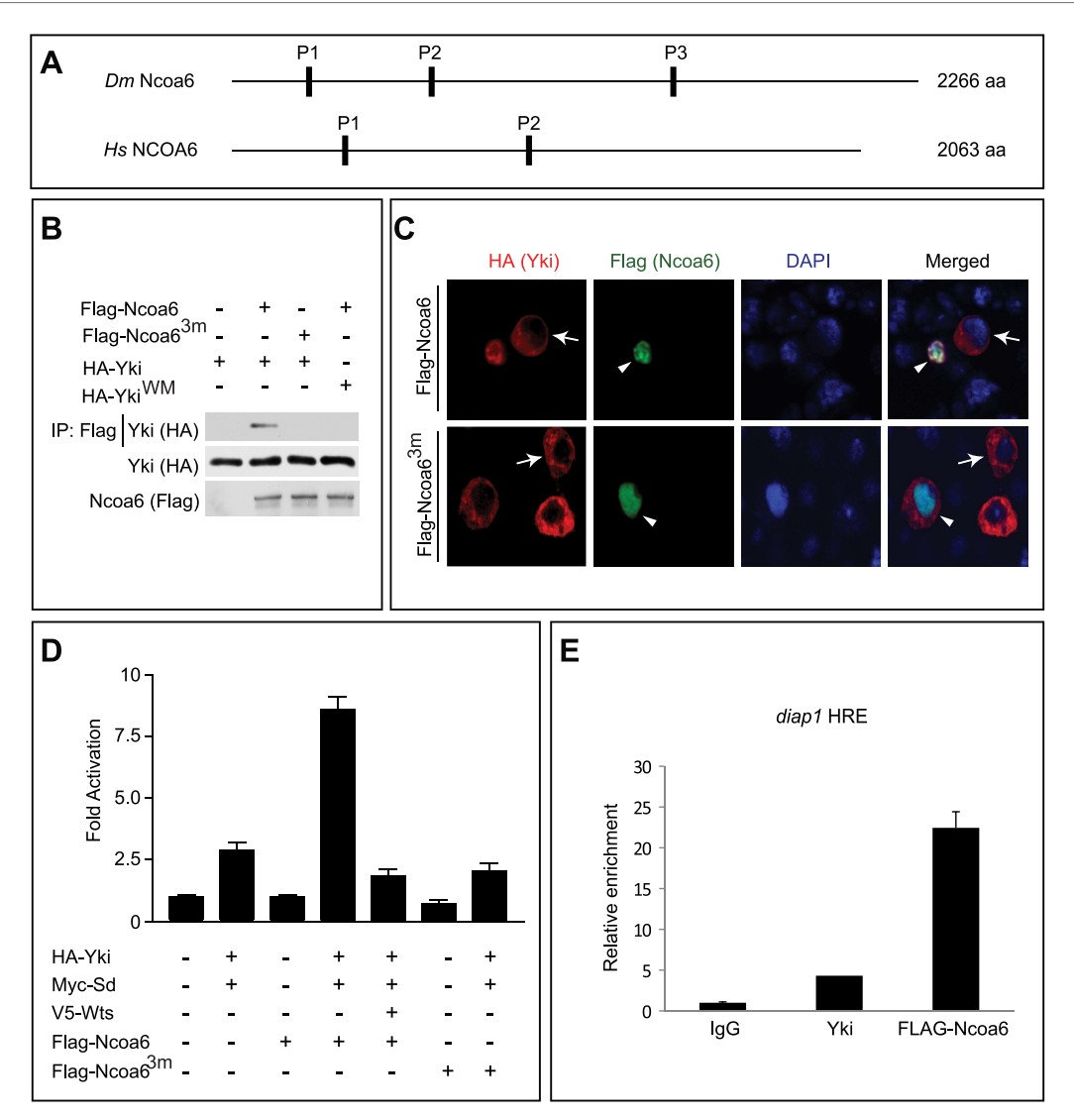

**Figure 1**. Ncoa6 physically interacts with Yki and regulates HRE activity. (**A**) Schematic protein structure of *Drosophila* Ncoa6 and its human orthologue, which contain three and two PPxY motifs, respectively. (**B**) S2R+ cells expressing the indicated constructs were subjected to immunoprecipitation as indicated. Note the physical interactions between Ncoa6 and Yki, and absence of interactions between Ncoa6³ᵐ and Yki or between Ncoa6 and Yki$^{WM}$. (**C**) *Drosophila* S2R+ cells co-transfected with HA-Yki and FLAG-Ncoa6 or FLAG-Ncoa6³ᵐ constructs were stained for the indicated epitopes. Cells with or without FLAG expression are marked by arrowheads and arrows, respectively. Both FLAG-Ncoa6 and FLAG-Ncoa6³ᵐ were localized to the nucleus (arrowheads), while HA-Yki was more concentrated in the cytoplasm (arrows). FLAG-Ncoa6, but not FLAG-Ncoa6³ᵐ, induced nuclear accumulation of HA-Yki (compare arrowheads in the merged channel). (**D**) Luciferase activity was measured in triplicates in *Drosophila* S2R+ cells transfected with the indicated constructs. Ncoa6, but not Ncoa6³ᵐ, enhanced Yki/Sd-mediated activation of HRE-luciferase reporter. This enhancement was suppressed by co-expression of Wts. Error bars represent standard deviations. (**E**) *Drosophila* S2R+ cells expressing FLAG-tagged Ncoa6 were subjected to ChIP analysis using control IgG, antibodies against FLAG or antibodies against endogenous Yki. The enrichment of HRE at the endogenous *diap1* locus was measured by real-time PCR. Both Yki and FLAG-Ncoa6 bound to the *diap1* HRE.

The following figure supplement is available for figure 1:

**Figure supplement1**. Identification of Ncoa6 as a positive regulator of the HRE activity from cell-based RNAi screen.

WW domains. Since Yki contains two WW domains, we hypothesized that Ncoa6 may potentiate Yki-mediated transcriptional activation through physical interactions with Yki. Indeed, epitope-tagged Ncoa6 and Yki coimmunoprecipitated with each other in *Drosophila* S2R+ cells (*Figure 1B*). This interaction was abolished by mutating the three PPxY motifs in Ncoa6 (Ncoa6[3m]) or the two WW domains in Yki (Yki[WM]) (*Figure 1B*), suggesting that the Ncoa6–Yki interaction was mediated by Ncoa6's PPxY motifs and Yki's WW domains. In agreement with this conclusion, we found that wild-type Ncoa6, but not Ncoa6[3m], promoted nuclear accumulation of Yki in S2R+ cells in co-transfection assays (*Figure 1C*). Of note, a recently published Hippo pathway protein–protein interactome included Ncoa6 as one of 245 proteins that were co-immunopreciated by Yki (*Kwon et al., 2013*).

Consistent with our observation that RNAi knockdown of Ncoa6 reduced the HRE reporter activity, overexpression of Ncoa6, but not the PPxY mutant Ncoa6[3m], potently enhanced Yki/Sd-mediated HRE reporter activity in *Drosophila* S2R+ cells (*Figure 1D*). In addition, the enhancement of HRE reporter activity by Ncoa6 was significantly suppressed by co-expression of the kinase Wts (*Figure 1D*). These results further support the importance of Ncoa6–Yki interactions in Hippo-responsive transcriptional regulation. Consistent with this notion, chromatin immunoprecipitation (ChIP) revealed that Ncoa6, like Yki, binds to the HRE site in the endogenous *diap1* gene locus in S2R+ cells (*Figure 1E*).

Since mutant alleles of Ncoa6 are not available, we used a previously validated transgenic RNAi line (*Herz et al., 2012*) to assess the role of Ncoa6 in tissue growth and Hippo target expression *in vivo*. Expression of UAS-Ncoa6 RNAi by the *dpp*-Gal4 driver resulted in a significant decrease in the width of the *dpp*-expression domain in adult wings, which corresponds to the region bordered by veins L3 and L4 (*Figure 2A*). Examination of third instar larval wing imaginal discs revealed a corresponding decrease in the expression of *diap1* and *four-jointed (fj)*, two well-characterized Hippo pathway target genes (*Figure 2B–C,E–F*). These results suggest that Ncoa6 is required for normal tissue growth and expression of Hippo target genes *in vivo*.

Next, we examined genetic interactions between Ncoa6 and the Hippo pathway. Overexpression of Yki or RNAi of Wts by the GMR-Gal4 driver leads to increased eye size (*Figure 3D,G*). Both phenotypes were suppressed by RNAi knockdown of Ncoa6 (*Figure 3E,H*). Conversely, knockdown of Ncoa6 exacerbated the small eye size induced by Sd overexpression (*Figure 3J–K*). To further investigate the genetic interactions between Ncoa6 and the Hippo pathway, we used Mosaic Analysis with a Repressible Cell Marker (MARCM) (*Lee and Luo, 1999*) to examine the requirement of Ncoa6 in *hpo* mutant clones. Ncoa6 knockdown suppressed the overgrowth as well as the elevated Diap1 expression in *hpo* mutant clones (*Figure 4A–D*). In fact, *hpo* mutant clones with Ncoa6 knockdown showed a decrease in Diap1 expression, similar to wildtype clones with Ncoa6 knockdown (*Figure 4C–D*). These findings further implicate Ncoa6 in Hippo-mediated growth control and gene expression.

The physical interactions between Yki and Ncoa6, together with the requirement for Ncoa6 in tissue growth and Hippo target gene expression, suggest that Yki may function as a transcriptional coactivator by interacting with Ncoa6. Since Sd is the primary DNA-binding transcription partner for Yki (*Koontz et al., 2013*), we reasoned that fusing the DNA-binding domain of Sd with Ncoa6 may directly target Ncoa6 to Hippo target genes and therefore stimulate their transcription in a Yki-independent manner (i.e., bypassing the genetic requirement for Yki). We tested this hypothesis in *Drosophila* wing discs using the MARCM technique. As reported before (*Huang et al., 2005*), *yki* mutant clones grew poorly and the rarely recovered clones always showed decreased Diap1 levels (*Figure 5A–B*). MARCM clones expressing a fusion protein between the DNA-binding domain of Sd and Ncoa6 (SdDB-Ncoa6) resulted in rounded clone morphology and dramatically increased Diap1 levels (*Figure 5C*). Significantly, the SdDB-Ncoa6 fusion protein, but not wild-type Ncoa6, rescued the growth defect and the decreased Diap1 levels in *yki* mutant clone (*Figure 5D–E*). In fact, *yki* mutant clones with SdDB-Ncoa6 overexpression were indistinguishable in clone size and Diap1 expression compared to wild-type clones with SdDB-Ncoa6 overexpression (*Figure 5C–D*). Thus, the SdDB–Ncoa6 fusion protein exhibits gain-of-function activity in a Yki-independent manner. Consistent with this notion, the SdDB-Ncoa6 fusion protein robustly stimulated the HRE-luciferase reporter in S2R+ cells in a manner that was not suppressed by co-expression of Wts (*Figure 5F*), in contrast to Wts' ability to suppress the HRE-luciferase reporter activity stimulated by the co-transfection of Sd, Yki, and Ncoa6 (*Figure 1D*).

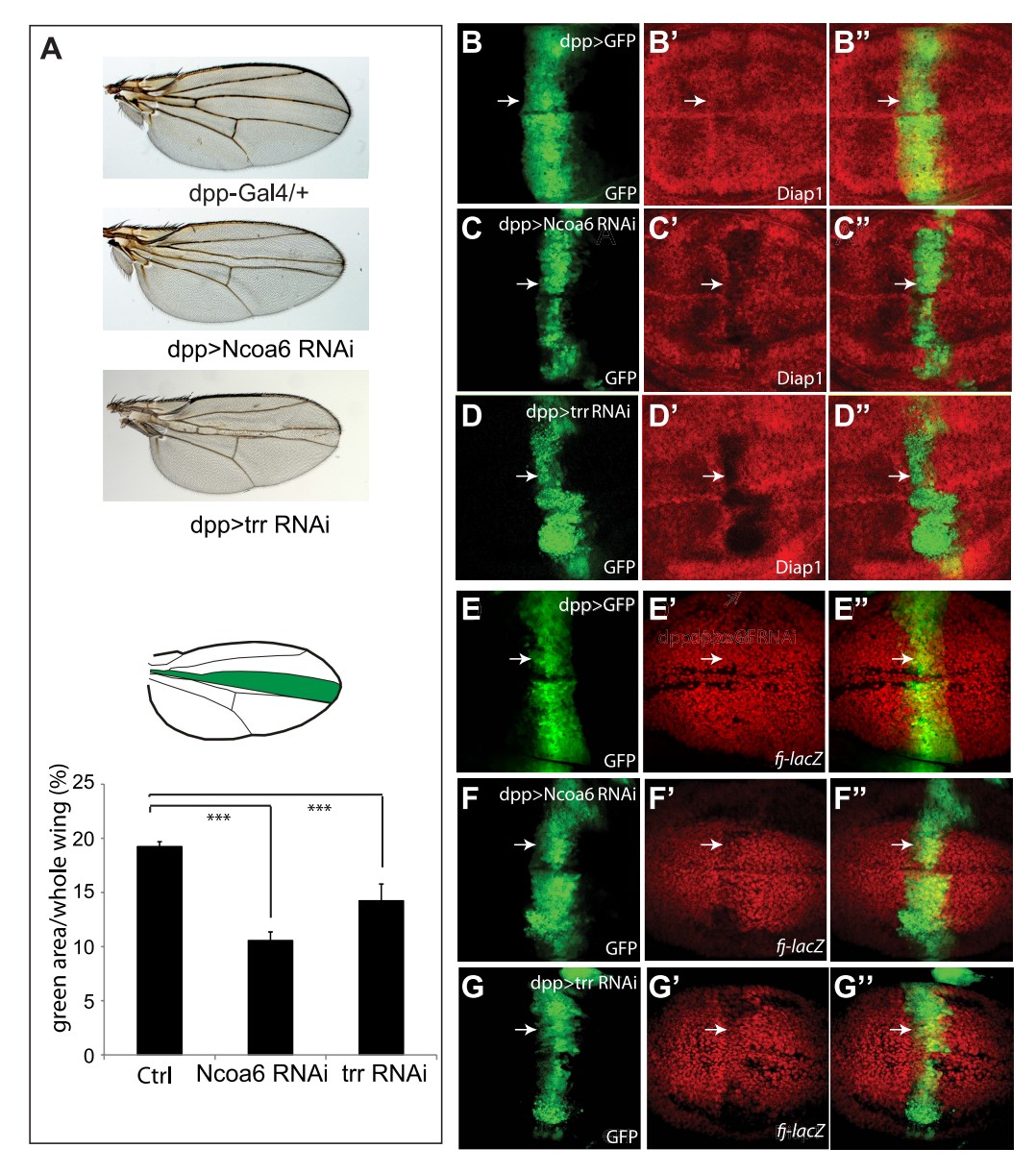

**Figure 2**. Ncoa6 and Trr are required for normal tissue growth and expression of Hippo target genes in *Drosophila* imaginal discs. (**A**) RNAi knockdown of Ncoa6 and Trr by *dpp*-Gal4 resulted in decreased area of the *dpp* expression domain in adult wings. The pictures were taken under the same magnification. The graph shows quantification of the *dpp* expression domain (green area in the schematic drawing) relative to the entire wing area (mean ± SEM, n = 14, ***p<0.001). The complete genotypes are: UAS-Dicer2; *dpp*-Gal4 UAS-GFP (control), UAS-Dicer2; *dpp*-Gal4 UAS-GFP/ UAS-Ncoa6RNAi (Ncoa6 RNAi), and UAS-Dicer2; *dpp*-Gal4 UAS-GFP/UAS-trrRNAi (trr RNAi). (**B–G**) RNAi knockdown of Ncoa6 or Trr resulted in decreased expression of Hippo target genes. Wing discs expressing UAS-GFP only (**B** and **E**), UAS-GFP plus Ncoa6 RNAi (**C** and **F**), or UAS-GFP plus trr RNAi (**D** and **G**) were stained for Diap1 (**B–D**) or *fj-lacZ* (**E–G**). Note the reduction of Diap1 and *fj-lacZ* levels upon Ncoa6 or Trr RNAi. The complete genotypes are: UAS-Dicer2; *dpp*-Gal4 UAS-GFP (**B**), UAS-Dicer2; *dpp*-Gal4 UAS-GFP/UAS-Ncoa6RNAi (**C**), UAS-Dicer; *dpp-Gal4* UAS-GFP/UAS-trr RNAi (**D**), UAS-Dicer2; *fj-lacZ*; *dpp*-Gal4 UAS-GFP (**E**), UAS-Dicer2; *fj-lacZ*; *dpp*-Gal4 UAS-GFP/UAS-Ncoa6 RNAi (**F**), and UAS-Dicer2; *fj-lacZ*; *dpp*-Gal4 UAS-GFP/ UAS-trr RNAi (**G**).

The results presented above suggest that Yki activates gene expression, at least in part, by recruiting Ncoa6. Since Ncoa6 has been reported to be a specific subunit of the Trr methyltransferase complex, we examined whether Trr, the catalytic subunit of this methyltransferase complex, is also required for Hippo-mediated growth control and gene expression. Similar to Ncoa6, expression of UAS-trr RNAi by the *dpp*-Gal4 driver resulted in a significant decrease in the width of the *dpp*-expression domain

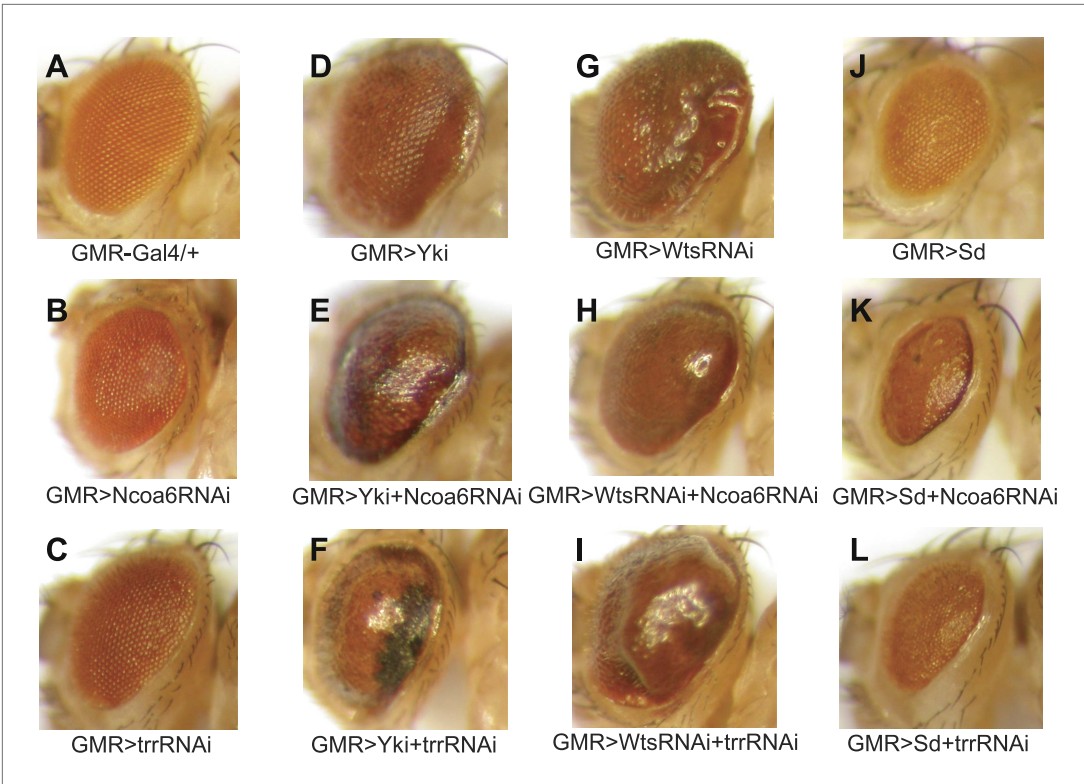

**Figure 3**. Genetic interactions between Ncoa6-Trr and the Hippo pathway. Adult eye images of the indicated genotypes, all taken under the same magnification. (**A**) GMR-Gal4/+. Wild-type control. (**B**) GMR-Gal4/+; UAS-Ncoa6 RNAi/+. RNAi knockdown of Ncoa6 resulted in a mild decrease in eye size (compare **B** to **A**). (**C**) UAS-Dicer2/+; GMR-Gal4/+; UAS-trr RNAi/+. RNAi knockdown of Trr resulted in no visible effects on eye size (compare **C** to **A**). (**D**) GMR-Gal4 UAS-Yki/+. Overexpression of Yki resulted in an increase in eye size (compare **D** to **A**). (**E**) GMR-Gal4 UAS-Yki/+; UAS-Ncoa6 RNAi/+. RNAi knockdown of Ncoa6 suppressed eye overgrowth induced by Yki overexpression (compare **E** to **D**). (**F**) UAS-Dicer2/+; GMR-Gal4 UAS-Yki/+; UAS-trr RNAi/+. RNAi knockdown of Trr suppressed eye overgrowth induced by Yki overexpression (compare **F** to **D**). (**G**) UAS-Wts RNAi/+; GMR-Gal4/+. RNAi knockdown of Wts resulted in an increase in eye size (compare **G** to **A**). (**H**) UAS-Wts RNAi/+; GMR-Gal4/ UAS-Ncoa6 RNAi. RNAi knockdown of Ncoa6 suppressed eye overgrowth induced by Wts knockdown (compare **H** to **G**). (**I**) UAS-Dicer2/+; UAS-Wts RNAi/+; GMR-Gal4/ UAS-trr RNAi. RNAi knockdown of Trr did not obviously suppress eye overgrowth caused by Wts knockdown. (**J**) GMR-Gal4 UAS-Sd/+. Overexpression of Sd resulted in a decrease in eye size (compare **J** to **A**). (**K**) GMR-Gal4 UAS-Sd/+; UAS-Ncoa6 RNAi/+. RNAi knockdown of Ncoa6 enhanced the small eye phenotype caused by Sd overexpression (compare **K** to **J**). (**L**) UAS-Dicer2/+; GMR-Gal4 UAS-Sd/+; UAS-trr RNAi/+. RNAi knockdown of Trr enhanced the small eye phenotype caused by Sd overexpression (compare **L** to **J**).

in adult wings and a corresponding decrease in the Hippo target genes *diap1* and *fj* (***Figure 2A,D,G***). Like Ncoa6, Trr knockdown suppressed eye overgrowth induced by Yki overexpression (***Figure 3D,F***) and aggravated the small eye phenotype caused by Sd overexpression (***Figure 3J,L***), although it did not visibly suppress eye overgrowth induced by Wts RNAi (***Figure 3I***). We also used MARCM to examine the requirement of Trr in *hpo* mutant clones. Similar to Ncoa6, Trr knockdown suppressed the overgrowth as well as the elevated Diap1 expression in *hpo* mutant clones (***Figure 4E–F***). Taken together, these results implicate the Trr methyltransferase complex in Hippo-mediated growth control and target gene expression.

The Trr methyltransferase complex in *Drosophila* mainly affects histone H3K4 monomethylation with subtle effect on H3K4 di- or trimethylation (***Herz et al., 2012***; ***Kanda et al., 2013***). To determine if Yki, Ncoa6, and Trr regulate growth in the Hippo pathway by modulating histone H3K4 methylation, we first examined the global levels of histone H3K4 methylation in *Drosophila* wing imaginal discs. It was reported previously that RNAi knockdown of Trr (using the *en*-Gal4 driver) led to a strong decrease

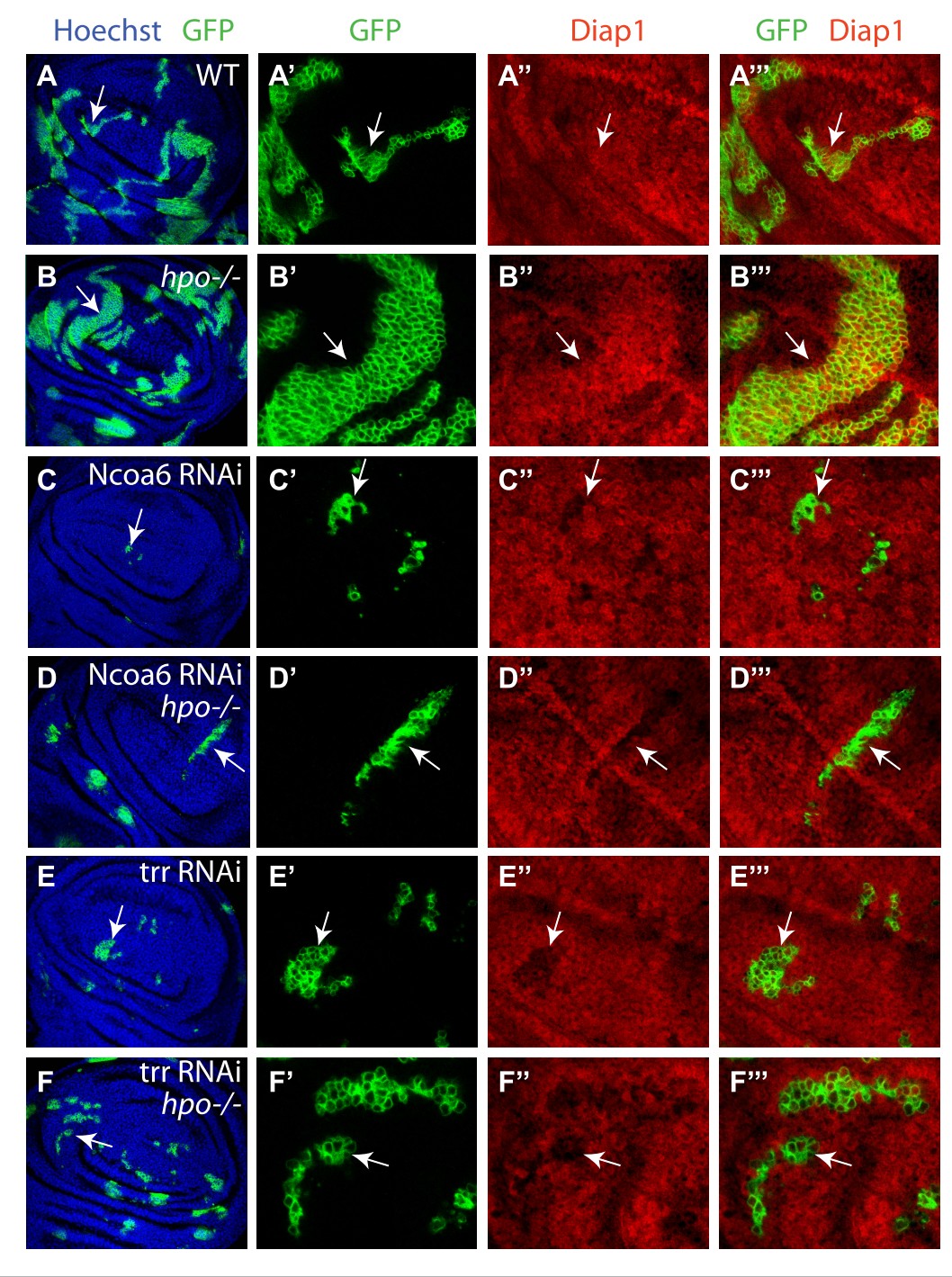

**Figure 4**. Ncoa6 and Trr are required for Hippo-mediated target gene expression. Wing discs containing GFP-marked MARCM clones were stained for Diap1 (red). For each genotype, the left most panel shows low magnification view of the wing disc (Hoechst + GFP), while the remaining three panels show higher magnification view of the same wing disc (GFP, Diap1 and GFP + Diap1). (A–F) Wing discs containing GFP-marked MARCM clones (green) of WT control (**A**), *hpo* mutant (**B**), Ncoa6 RNAi (**C**), *hpo* mutant with Ncoa6 RNAi (**D**), Trr RNAi (**E**), and *hpo* mutant with Trr RNAi (**F**). Note the increased Diap1 levels in *hpo* mutant clones and the decreased Diap1 levels in Ncoa6 RNAi or Trr RNAi clones. Also note the decreased Diap1 levels in *hpo* mutant clones with Ncoa6 RNAi or Trr RNAi.

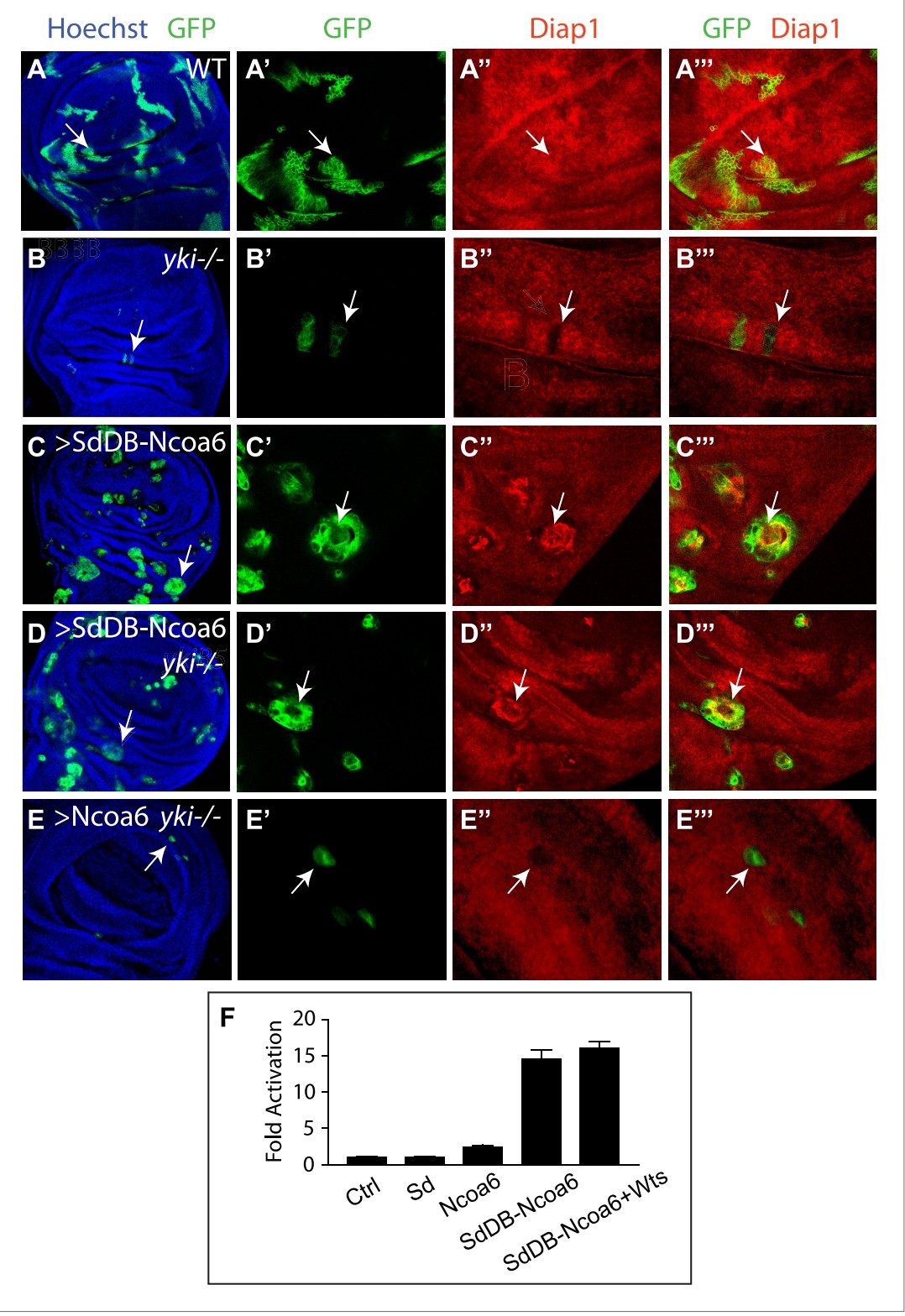

**Figure 5**. Fusion of Ncoa6 with the DNA binding domain of Sd bypasses Yki to stimulate Hippo target gene and tissue growth. (**A**–**E**) Wing discs containing GFP-marked MARCM clones (green) of WT control (**A**), $yki^{B5}$ (**B**), SdDB-Ncoa6 overexpression (**C**), $yki^{B5}$ with SdDB-Ncoa6 overexpression (**D**), and $yki^{B5}$ with Ncoa6 overexpression (**E**),were stained for Diap1 (red). For each genotype, the left most panel shows low magnification view of the wing

*Figure 5. Continued on next page*

*Figure 5. Continued*
disc (Hoechst + GFP), while the remaining three panels show higher magnification view of the same wing disc (GFP, Diap1, and GFP + Diap1). Note the decreased Diap1 expression and undergrowth of *yki^B5* clones (**B**) or *yki^B5* clones with Ncoa6 overexpression (**E**). SdDB-Ncoa6 overexpression resulted in elevated Diap1 levels in *yki^B5* clones (**D**). (**F**) Luciferase activity was measured in triplicates in *Drosophila* S2R+ cells transfected with the indicated constructs. Error bars represent standard deviations. Note the Wts-insensitive stimulation of the HRE-luciferase reporter by SdDB-Ncoa6.

in H3K4me1 in the posterior compartment of the wing imaginal discs, while it had marginal effects on H3K4me2 and H3K4me3 levels (*Mohan et al., 2011*; *Herz et al., 2012*). It was also showed that RNAi knockdown of Ncoa6 in the posterior compartment of the wing imaginal disc resulted in a weak reduction in H3K4me1 levels (*Herz et al., 2012*), which we confirmed (*Figure 6—figure supplement1*). We further examined H3K4me2 and H3K4me3 levels in these imaginal discs, and observed a very subtle decrease in H3K4me3 levels and no detectable changes in H3K4me2 levels (*Figure 6—figure supplement 1B–C*). However, when we examined mutant clones of *yki* in the wing imaginal discs, we could not detect any changes in the global levels of H3K4me1, H3K4me2, or H3K4me3 (*Figure 6—figure supplement1*).

Given the negligible effect of Yki on global levels of H3K4 methylation, we investigated whether Yki modulates local H3K4 methylation on Hippo target genes. It is well established that H3K4 mono-methylation marks enhancers and actively transcribed introns, while H3K4 trimethylation is enriched at active promoters and transcription start site (TSS)-proximal regions (*Heintzman et al., 2007*; *Kharchenko et al., 2011*). Interestingly, a previous genome-wide analysis in *Drosophila* S2 cells revealed that *diap1* and *ex*, two well-characterized Hippo target genes, display such differential enrichment of H3K4me1 and H3K4me3 at the respective region of each gene (*Herz et al., 2012*) (*Figure 6A*). To examine the contribution of Yki, Ncoa6, and Trr to H3K4 methylation at these Hippo target genes, we knocked down each protein in S2R+ cells and performed ChIP analysis with antibodies against H3K4me1 and H3K4me3. RNAi knockdown of Yki, Ncoa6 or Trr resulted in a decrease of H3K4me3 in the TSS-proximal region of *diap1* and *ex* (*Figure 6B*), which normally showed the strongest enrichment of H3K4me3 marks (*Herz et al., 2012*) (*Figure 6A*). RNAi knockdown of Yki, Ncoa6 or Trr also led to a decrease of H3K4me1 in the intronic HRE of *diap1* and an upstream region of *diap1* or *ex* which normally showed the strongest enrichment of H3K4me1 marks (*Herz et al., 2012*) (*Figure 6C*). Collectively, these data are consistent with the view that Yki activates target gene transcription by interacting with the Trr methyltransferase complex and modifying the chromatin state of the target loci.

Despite the ever expanding complexity of upstream inputs into the Hippo pathway, all of them converge on the transcriptional coactivator Yki. Thus, understanding the molecular mechanisms by which Yki regulates tissue growth and target gene expression has important implications for developmental and cancer biology. Previous studies have established that Yki functions primarily as a coactivator for Sd and that Yki promotes tissue growth by antagonizing Sd's repressor function (*Wu et al., 2008*; *Koontz et al., 2013*). Our current study has extended the previous work by identifying Ncoa6 as a Yki-binding cofactor that is required for the expression of Yki target genes. The ability of the SdDB-Ncoa6 fusion protein to rescue the growth and transcriptional defects in *yki* mutant clones highlights the importance of Ncoa6 recruitment in the transcriptional output of the Hippo pathway. Our results further suggest that Ncoa6 recruits the Trr methyltransferase complex to Hippo target genes and that Yki regulates target gene transcription by modulating local H3K4 methylation. Consistent with this view, a recent genome-wide chromatin-binding analysis revealed a correlation between Yki-bound chromatin and peaks of H3K4me3 modification in *Drosophila* wing discs and embryos (*Oh et al., 2013*). We note that, besides the H3K4 methyltransferases, the mammalian Ncoa6 has been reported to potentiate the activity of transcription factors by interacting with histone acetyltransferase CBP/p300 and several RNA binding proteins (CAPER, CoAA and PIMT) (*Mahajan and Samuels, 2008*). Whether these additional mechanisms also contribute to the function of Ncoa6 in Yki-mediated growth control requires further investigation. Given the biological and clinical significance of the Hippo pathway, further studies into the molecular mechanism of Ncoa6 will advance our understanding of developmental growth control and facilitate the development of novel therapeutic strategies.

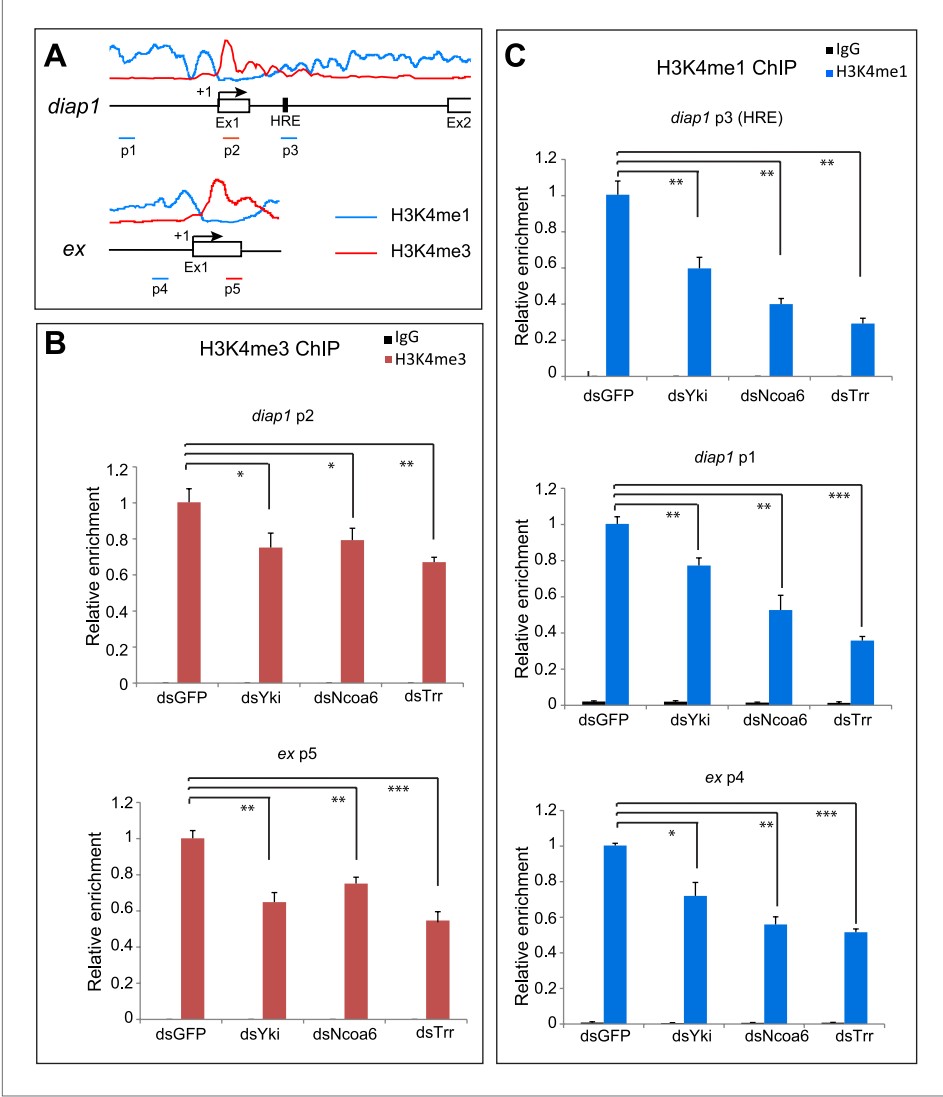

**Figure 6**. Yki modulates local H3K4 methylation at Hippo target genes. (**A**) Schematic view of *diap1* and *ex* genomic loci analyzed by ChIP. Transcriptional start site is labeled as +1, and p1–p5 are a series of primer sets encompassing following regions of *diap1* and *ex*: *diap1*: p1: −1951 ~ −1813, p2: +228 ~ +377, p3: +3993 ~ +4104; *ex*: p4: −749 ~ −608, p5: +249 ~ +393. Note that p3 covers the *diap1* HRE. Also shown are the profiles of H3K4me1 (blue line) and H3K4me3 (red line) binding derived from a previously published ChIP-Seq analysis in S2 cells (*Herz et al., 2012*). (**B** and **C**) RNAi knockdown of Yki, Ncoa6 or Trr resulted in decreased H3K4me3 (**B**) and H3K4me1 (**C**) modification on Hippo target genes. ChIP analysis of H3K4me1 or H3K4me3 were performed in *Drosophila* S2R+ cells treated with dsRNA of GFP (control), Yki, Ncoa6, or Trr. Chromatins were precipitated by control IgG or antibodies against H3K4me1 and H3K4me3. The enrichment of ChIP products on *diap* and *ex* was measured by real-time PCR using the indicated primers. ***p<0.001, **p<0.01,*p<0.05.

The following figure supplement is available for figure 6:

**Figure supplement1**. Ncoa6, but not Yki, regulates global levels of H3K4 methylation.

## Materials and methods

### Molecular cloning and mutagenesis

A full-length Ncoa6 cDNA corresponding to the BDGP annotated RD transcript was generated from mRNA of *Drosophila* third instar larvae, using SuperScript(R) III One-Step RT-PCR System with Platinum Taq High Fidelity (Life Technologies, Carlsbad, California). Mutations of PPxY motifs were generated

in Ncoa6 using the QuikChange II XL Site-Directed Mutagenesis Kit (Agilent, Santa Clara, CA), replacing tyrosine(Y) with alanine (A). To generate Sd-Ncoa6, Sd DNA binding domain was inserted to the N-terminal of Ncoa6. FLAG-tag was inserted to the N-terminal of Ncoa6, Ncoa6$^{3m}$, and Sd-Ncoa6 and cloned into the *attB*-UAS vector.

## Drosophila genetics

Flies with the following genotypes have been described previously: *yki*$^{B5}$, UAS-Yki (*Huang et al., 2005*), *hpo*$^{42-48}$ (*Wu et al., 2003*), *fj-lacZ* reporter *fj*$^{9-II}$ (*Villano and Katz, 1995*), UAS-Sd (*Halder et al., 1998*), UAS-*Wts* RNAi (Stock ID VDRC v106174). The UAS-*Ncoa6* RNAi and UAS-*trr* RNAi lines have been validated previously (*Herz et al., 2012*) and were obtained from Bloomington *Drosophila* Stock Center (Stock ID 34964 and 29,563). *attB*-UAS-Ncoa6 and *attB*-UAS-FLAG-SdDB-Ncoa6 transgenes were inserted into the 86Fa attP acceptor site by phiC31-mediated site-specific transformation (*Bischof et al., 2007*).

For the MARCM experiments in *Figure 4*, the following clones were induced 48–60 hr after egg deposition and heat shocked at 37°C for 15 min:

UAS-GFP hs-FLP; FRT42D, Tub-Gal80/FRT42D; Tub-Gal4/+
UAS-GFP hs-FLP; FRT42D, Tub-Gal80/FRT42D hpo$^{42-48}$; Tub-Gal4/+
UAS-GFP hs-FLP; FRT42D, Tub-Gal80/FRT42D; Tub-Gal4/UAS-Ncoa6RNAi
UAS-GFP hs-FLP; FRT42D, Tub-Gal80/FRT42D hpo$^{42-48}$; Tub-Gal4/UAS-Ncoa6RNAi
UAS-Dicer2/UAS-GFP hs-FLP; FRT42D, Tub-Gal80/FRT42D; Tub-Gal4/UAS-trrRNAi
UAS-Dicer2/UAS-GFP hs-FLP; FRT42D, Tub-Gal80/FRT42D hpo$^{42-48}$; Tub-Gal4/UAS-trrRNAi

For the MARCM experiments in *Figure 5A–E*, the following clones were induced 72–84 hr after egg deposition and heat shocked at 37°C for 10 min:

UAS-GFP hs-FLP; FRT42D, Tub-Gal80/FRT42D; Tub-Gal4/+
UAS-GFP hs-FLP; FRT42D, Tub-Gal80/FRT42D yki$^{B5}$; Tub-Gal4/+
UAS-GFP hs-FLP; FRT42D, Tub-Gal80/FRT42D; Tub-Gal4/UAS-SdDB-Ncoa6
UAS-GFP hs-FLP; FRT42D, Tub-Gal80/FRT42D yki$^{B5}$; Tub-Gal4/UAS-SdDB-Ncoa6
UAS-GFP hs-FLP; FRT42D, Tub-Gal80/FRT42D yki$^{B5}$; Tub-Gal4/UAS-Ncoa6

## Drosophila cell culture, transfection, immunoprecipitation, immunoflurescence, and luciferase reporter assay

*Drosophila* S2R+ cells were cultured in Schneider's *Drosophila* Medium (Life Technologies) supplemented with 10% fetal bovine serum and antibiotics. HA-Yki and HA-Yki$^{WM}$ have been described previously (*Huang et al., 2005*). Luciferase assay was carried out using Dual Luciferase Assay System (Promega, Madison, WI) and a FLUOstar Luminometer (BMG LabTechnologies, Germany). Tansfection, immunopreciptation, and immunofluorescence staining of S2R+ cells were performed using standard protocols as described (*Yin et al., 2013*).

## ChIP assays

ChIP assays were performed according to a previously described protocol (*Wang et al., 2009*). Briefly, ~5 × 10$^6$ (for ChIP assay with histone methylation antibodies) or 1.5 × 10$^7$ (for CHIP assay with Yki or FLAG antibodies) S2R+ cells were cross-linked with 1% formaldehyde and sonicated to an average fragment size between 200 bp and 500 bp. Two micrograms of control IgG or specific antibodies, including rabbit α-H3K4me1 (8895, Abcam, England) and rabbit α-H3K4me3 (8580, Abcam), and 50 μl of protein G agarose were used in each ChIP assay. The immunoprecipitated DNA was quantified using real-time PCR. All values were normalized to the input. The primers for analyzing the ChIP NA are provided as follows:

p1 Forward: TGTTCTTGTTGGTGCTGCTT
p1 Reverse: TTAATGCTGGCATGGTTTCA
p2 Forward: TAAAACTGGGGCTCACCTTG
p2 Reverse: TCGTGTTCACGGAAAATCAA
p3 (HRE) Forward: ACGAACACGAAGACCAAA
p3 (HRE) Reverse: CTCCAAGCCAGTTTGATT
p4 Forward: AAAAGAGGGAAGAGGGAGCA

p4 Reverse: GAATCGGAATCGGAACTTGA
p5 Forward: TCGCACTCGCCTCAATTAC
p5 Reverse: CAGCACCAACTTTTCGGAGT

## Acknowledgements

We thank Jianzhong Yu, Melissa Jones and Elizabeth Garcia for technical assistance. This study was supported in part by grants from the National Institutes of Health (EY015708). DP is an investigator of the Howard Hughes Medical Institute.

## Additional information

### Competing interests

DP: Reviewing editor, *eLife*. The other authors declare that no competing interests exist.

### Funding

| Funder | Grant reference number | Author |
| --- | --- | --- |
| Howard Hughes Medical Institute | | Yun Qing, Feng Yin, Wei Wang, Yonggang Zheng, Pengfei Guo, Frederick Schozer, Hua Deng, Duojia Pan |
| National Institutes of Health | NIH EY015708 | Yun Qing, Feng Yin, Wei Wang, Yonggang Zheng, Pengfei Guo, Frederick Schozer, Hua Deng, Duojia Pan |

The funders had no role in study design, data collection and interpretation, or the decision to submit the work for publication.

### Author contributions

YQ, FY, Conception and design, Acquisition of data, Analysis and interpretation of data, Drafting or revising the article; WW, YZ, Conception and design, Acquisition of data, Analysis and interpretation of data; PG, HD, Acquisition of data; FS, Acquisition of data, Analysis and interpretation of data; DP, Conception and design, Analysis and interpretation of data, Drafting or revising the article

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
