## [Decision Letter]

Thank you for sending your work entitled ”The Hippo signaling effector Yorkie regulates target gene transcription by recruiting Nuclear receptor coactivator 6” for consideration at *eLife*. Your article has been favorably evaluated by Senior editor Janet Rossant and 2 reviewers, one of whom is a member of our Board of Reviewing Editors.

The Senior editor and the two reviewers discussed their comments before we reached this decision, and the Senior editor has assembled the following comments to help you prepare a revised submission.

In this manuscript, the authors identify Ncoa6, a subunit of the H3K4 methyltransferase Trr complex, as a novel Yorkie-interacting protein. Given how Yorkie functions as a transcriptional co-activator is not well understood, this manuscript provides an important advance in the field. Thus, the potential impact is high.

However, although the authors demonstrate that Ncoa6 is required for normal growth and Hippo target gene expression, in vivo evidence that Trr complex are functionally required for Hippo-mediated growth and gene expression is completely lacking in this manuscript. It would be important to examine whether knock-down of Ncoa6 and Trr suppress (i) eye over-growth (in Figure 3) and (ii) elevation of Diap1 expression in the wing disc (in Figure 4), in hippo or warts mutant clones/tissue. We believe that performing the additional above experiments would significantly improve the manuscript.

Other issues to address:

*Reviewer #1*:

As a minor point, the authors mention that FLAG-Ncoa6 is localized in the nucleus. How can Ncoa6 that is only present in the nucleus promote nuclear translocation of HA-Yorkie that is present in the cytoplasm?

*Reviewer #2*:

Figure 1: The authors show nicely that Flag-Ncoa6 binds to Yki in IP experiments, and that the interaction depends on the motifs of Ncoa6. Ncoa6 addition increases the ability of Yki to promote activation of a transcriptional reporter, and to be enriched at the diap1 HRE. Ncoa6 is in the nucleus at all times, and overexpression of Ncoa6 with a mutation in the PPxY motifs causes Yki to be largely localized in the cytoplasm.

What is the localization of Yki in this system without ectopic Ncoa6? One point that is a bit unclear is if endogenous Yki is localized to the nucleus based on Ncoa6 interactions. The authors might consider knocking down Ncoa6 with RNAi, and seeing if it causes Yki relocalization.

Figure 2: The authors nicely show that knockdown of Ncoa6 with RNAi via dppGal4 leads to a suppression of growth in the adult wing, and a reduction of the dpp domain in the larval wing disc. The authors also show that some Hippo tragets are affected by Ncoa6 RNAi. Diap1 and Fj-lacZ. The paper would be strengthened by adding another Hpo pathway readout-for example Ex protein staining, as they later show that Ex is a target of Ncoa6.

One puzzling aspect the authors might comment on is that there seems to be a degree of non-autonomy. For example in Fig E, the edges of the dpp-stripe expressed Ncoa6 RNAi do not show clear downregulation of Fj.

Figure 3: Genetic interaction screens in the eye support a role for Ncoa6 in growth control via the Hippo pathway.

Figure 4: Here the authors show that overexpression of Ncoa6 gives rise to tissue growth, while there is reduced growth when this is done in the absence of Yki. They create a fusion protein of Sd DNA binding domain and Ncoa6, which allows growth in the absence of Yki. Support for this growth effect being through the Hippo pathway is that a Hippo target, Diap1, is upregulated by the Sd-Ncoa6 fusion. Overexpression with dpp-Gal4 broadens this domain, consistent with an activation of growth.

I would appreciate if the authors would comment on the edge effect seen in panels E and F. It appears that there is a decrease in Diap1 staining at the edges of the clone, and strong upregulation inside the clones.

Figure 5: Ncoa6 is a subunit of the trithorax-related (Trr) methyltranserase complex. The show decreases in H3K4me1 ChIP at the diap1 p3 HRE with knockdown of Trr, Ncoa6 or Yki, also at ex p4 and diap p1. Similarly there is a decrease in H3K4me3. This is all consistent with the author's model. While not necessary for publication, it would be interesting to see if this effect is specific for this methyltransferase complex, or if dSet1 loss has similar effects.

---

## [Author Response]

*[…] However, although the authors demonstrate that Ncoa6 is required for normal growth and Hippo target gene expression, in vivo evidence that Trr complex are functionally required for Hippo-mediated growth and gene expression is completely lacking in this manuscript. It would be important to examine whether knock-down of Ncoa6 and Trr suppress (i) eye over-growth (in*
Figure 3*) and (ii) elevation of Diap1 expression in the wing disc (in*
Figure 4*), in hippo or warts mutant clones/tissue. We believe that performing the additional above experiments would significantly improve the manuscript*.

Thank you for the suggestions. In this revision, we have conducted additional experiments to investigate the role of the Trr complex in normal growth and Hippo target gene expression. Specifically, we have added data showing that 1) knockdown of Ncoa6 and Trr suppresses eye overgrowth caused by Yki overexpression (Figure 3); 2) knockdown of Ncoa6 suppresses eye overgrowth caused by Wts knockdown (Figure 3); 3) knockdown of Ncoa6 and Trr suppresses Diap1 expression in *hpo* mutant wing clones (Figure 4). Accordingly, we have slightly modified the title of the manuscript in the revision. We hope that these additional experiments have significantly improved our manuscript.

*Other issues to address*:

Reviewer #1:

*As a minor point, the authors mention that FLAG-Ncoa6 is localized in the nucleus*. *How can Ncoa6 that is only present in the nucleus promote nuclear translocation of HA-Yorkie that is present in the cytoplasm?*

Yorkie, like many other phospho-regulated transcription factors in the cells, shuttles between the nucleus and the cytoplasm. Thus, its steady-state subcellular localization in a cell reflects the equilibrium of nuclear import/retention and cytoplasmic export/retention. Any conditions that increase nuclear retention, as by the overexpression of the nuclear FLAG-Ncoa6 protein, would shift the equilibrium to the nucleus. Conversely, conditions that increase cytoplasmic retention, as by the binding of the cytoplasmic protein 14-3-3, would shift the localization towards the cytoplasm. In order to better convey the point that the snapshot of Yki localization in our co-transfection experiment is the equilibrium of a dynamic process, we use “nuclear accumulation” instead of “nuclear translocation” in the revised manuscript.

Reviewer #2:

Figure 1*: The authors show nicely that Flag-Ncoa6 binds to Yki in IP experiments, and that the interaction depends on the motifs of Ncoa6. Ncoa6 addition increases the ability of Yki to promote activation of a transcriptional reporter, and to be enriched at the diap1 HRE. Ncoa6 is in the nucleus at all times, and overexpression of Ncoa6 with a mutation in the PPxY motifs causes Yki to be largely localized in the cytoplasm*.

*What is the localization of Yki in this system without ectopic Ncoa6? One point that is a bit unclear is if endogenous Yki is localized to the nucleus based on Ncoa6 interactions. The authors might consider knocking down Ncoa6 with RNAi, and seeing if it causes Yki relocalization*.

As shown in Figure 1, HA-Yki was mostly cytoplasmic in S2R+ cells without ectopic Ncoa6, but was mostly nuclear in the presence of ectopic Ncoa6, and an Ncoa6 protein with PPxY mutations had no effect on HA-Yki localization. Following the reviewer’s suggestion, we examined whether knocking down Ncoa6 affects Yki localization and the results were mixed. We found that Ncoa6 knockdown in S2R+ cells led to decreased nuclear localization of HA-Yki, although it had no obvious effect on endogenous Yki. This data is included for your perusal at the end of this letter (Figure A–B). I should emphasize that the results presented in Figure 1 were intended to further support the physical interactions between Yki and Ncoa6 – they were not intended to imply that endogenous Yki is localized to the nucleus based on Ncoa6 interactions.

Figure 2*: The authors nicely show that knockdown of Ncoa6 with RNAi via dppGal4 leads to a suppression of growth in the adult wing, and a reduction of the dpp domain in the larval wing disc. The authors also show that some Hippo tragets are affected by Ncoa6 RNAi. Diap1 and Fj-lacZ. The paper would be strengthened by adding another Hpo pathway readout-for example Ex protein staining, as they later show that Ex is a target of Ncoa6*.

*One puzzling aspect the authors might comment on is that there seems to be a degree of non-autonomy. For example in Figure E, the edges of the dpp-stripe expressed Ncoa6 RNAi do not show clear downregulation of Fj*.

We have examined Ex staining but saw no change upon Ncoa6 RNAi. These data are included as Figure 7 below. We do not know the exact reason but it could be due to the efficiency of *in vivo* RNAi by UAS-RNAi transgene.Author response image 1.(A) Cell fractionation of S2R+ cells showing that RNAi knockdown of Ncoa6 modestly decreased HA-Yki in the nucleus, with a concurrent increase in the cytoplasm.(B) RNAi knockdown of Ncoa6 caused no obvious change in subcellular fractionation of endogenous Yki. Sd, which has been reported to promote Yki nuclear localization, was included as a control. Knockdown of Sd led to a modest decrease of nuclear Yki. (C-D) RNAi knockdown of Ncoa6 did not obviously affect the expression of Ex. Wing discs expressing UAS-GFP only (C) or UAS-GFP plus Ncoa6 RNAi (D) were stained for Ex. The complete genotypes are: UAS-Dicer2; *dpp*-Gal4 UAS-GFP (C), UAS-Dicer2; *dpp*-Gal4 UAS- GFP/UAS-Ncoa6RNAi (D).

The reviewer noted that the edges of the *dpp*-stripe expressing Ncoa6 RNAi do not show clear downregulation of Fj. We think this is likely due to the fact that the expression level of the *dpp*-Gal4 is not uniform across the stripe – it is strongest in the center of the stripe and tapers off towards the two edges (please see GFP staining in Figure 2).

Figure 3*: Genetic interaction screens in the eye support a role for Ncoa6 in growth control via the Hippo pathway*.

Figure 4*: Here the authors show that overexpression of Ncoa6 gives rise to tissue growth, while there is reduced growth when this is done in the absence of Yki. They create a fusion protein of Sd DNA binding domain and Ncoa6, which allows growth in the absence of Yki. Support for this growth effect being through the Hippo pathway is that a Hippo target, Diap1, is upregulated by the Sd-Ncoa6 fusion. Overexpression with dpp-Gal4 broadens this domain, consistent with an activation of growth*.

*I would appreciate if the authors would comment on the edge effect seen in panels E and F. It appears that there is a decrease in Diap1 staining at the edges of the clone, and strong upregulation inside the clones*.

We were puzzled by this as well but could only speculate on the reasons for edge effect in these clones. One possibility is the induction of an unknown protein in the mutant clones by the Sd-Ncoa6 fusion protein that leads to a non-cell autonomous effect at the clone boundary (such as the Ft/Ds/Fj system).

Figure 5*: Ncoa6 is a subunit of the trithorax-related (Trr) methyltranserase complex. The show decreases in H3K4me1 ChIP at the diap1 p3 HRE with knockdown of Trr, Ncoa6 or Yki, also at ex p4 and diap p1. Similarly there is a decrease in H3K4me3. This is all consistent with the author's model. While not necessary for publication, it would be interesting to see if this effect is specific for this methyltransferase complex, or if dSet1 loss has similar effects*.

There are three H3K4 methyltransferase complexes in *Drosophila*, trithorax, trithorax-related, and dSet1. We have only focused on the Trr complex due to its physical interactions with Yki through Ncoa6 binding. We agree with the reviewer in that examining the other two methyltransferase complexes is best left for future studies. Furthermore, unlike the Trr complex, the other two methyltransferase complexes were not identified in any of the protein interaction or reporter-based screens.